# ATTRIBUTE-GUIDED DIFFUSION FOR UNSUPERVISED FEW-SHOT FONT GENERATION

## ABSTRACT

Font generation is a challenging problem, especially for some writing systems composed of a large number of characters, such as Chinese, which has attracted the attention of many scholars in recent years. However, existing font generation methods are usually based on generative adversarial networks. Due to the problems of training instability and mode collapse in generative adversarial networks, the performance of many methods has encountered bottlenecks. In order to solve this problem, we apply the latest generative model — the diffusion model to this task. We use the method of decoupling content and style to extract image attributes, combine the required content and style with the input diffusion model as a condition, and then guide diffusion models to generate glyphs corresponding to styles. Our method can be stably trained on large datasets and our model achieves pretty good performance both qualitatively and quantitatively compared with previous font generation methods.

## 1 INTRODUCTION

In this era, people read a lot of text on the Internet every day. Text is also an important tool for information transmission and storage. In order to make the text look interesting, many fonts were created and used. The fonts have generated enormous commercial value. However, traditional font design methods rely heavily on professional designers who need to draw each character individually, which is particularly tedious and time-consuming, especially in languages with rich characters and complex structures, such as Chinese ($> 60,000$ characters), Japanese ($> 50,000$ characters), and Korean ($> 11,000$ characters).

In recent years, with the development of convolutional neural networks, computers have gradually been able to automatically generate high-quality real pictures. Therefore, the few-shot font generation (FFG) task emerges, which aims to generate a complete font library from only a few reference font images. An ideal FFG system can greatly reduce the burden of the time-consuming and labor-intensive font design process. This task has attracted the interest of many researchers and much progress has been made in this field.

Recently, several attempts have been made with few-shot font generation with promising results. "Rewrite"(Rewrite) and "Zi2zi"(Zi2zi) generate characters by learning the mapping of thousands of paired characters from one style to another, but neither of them can generalize to unseen styles. To generate fonts not seen during training, EMD(Zhang et al., 2018b) designed neural networks to separate content and style representations, but the results were not ideal. Since then, results from many methods(Wu et al., 2020; Cha et al., 2020; Park et al., 2021a; Xie et al., 2021; Kong et al., 2022) have shown that content and style disentanglement are effective for font generation tasks. CalliGAN(Wu et al., 2020) generates learned embeddings of glyph images based on component labels and style labels, and thus cannot generalize to unseen styles or unseen components. DM-Font(Cha et al., 2020) uses a dual-memory architecture to generate fonts, but it needs a reference set containing all components to extract stored information, which is very expensive. LF-Font(Park et al., 2021a) can be extended to unseen styles according to component style features, however, its visual quality is significantly degraded in few-shot generation scenarios. DG-Font(Xie et al., 2021) realizes unsupervised font generation and designs deformable skip connections for better performance. CG-GAN(Kong et al., 2022) proposes a more fine-grained discriminator to supervise the generator to decouple content and style at a fine-grained level.

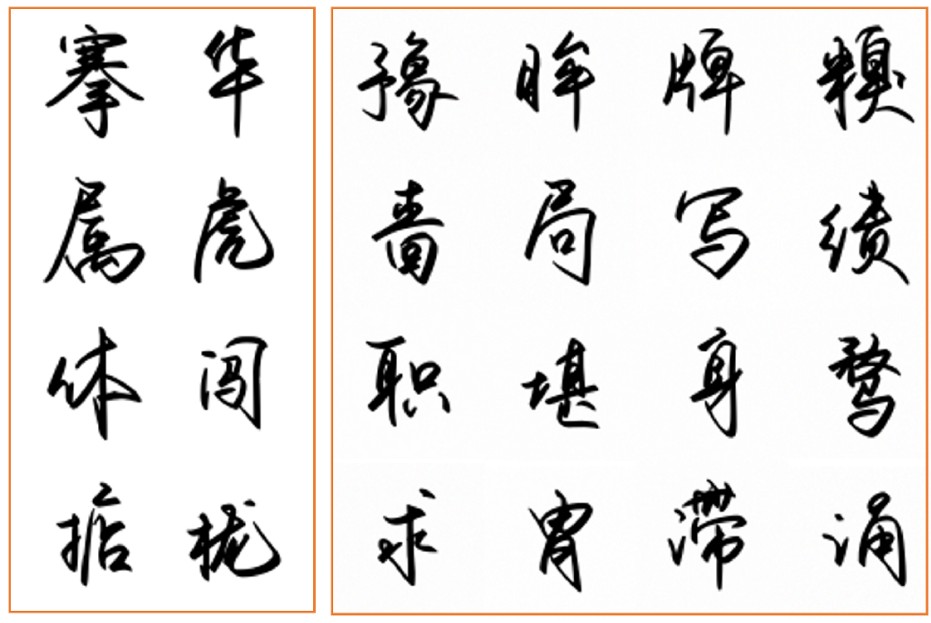

Reference Calligraphy          Imitation Result

Figure 1: **Unsupervised font generation results.** For this type of font with a more elegant style, the glyphs generated by our model also have rich details, such as stroke tips, joined-up writing, and thickness of strokes.

So far, most of the work is based on the generative adversarial networks(Goodfellow et al., 2020), which often requires careful design of hyperparameters to be able to train well, and the generated results often have the phenomenon of mode collapse. Limited by these shortcomings, these methods based on generative adversarial networks have reached a bottleneck.

In this paper, we apply the diffusion model to the task of few-shot font generation. Our model encodes two attributes of characters: content and style, and uses the diffusion model as a generator. Unlike image-to-image translation frameworks, our method does not learn the mapping between source and target font domains. We treat the content and style of characters as conditional variables fed into the diffusion model as a guide. The model learns how to use these conditional variables to generate the corresponding characters.

In summary, our main contributions include:

- We apply the latest generation model — diffusion models to the task of few-shot font generation. It can be stably trained on large data sets. This is a rare attempt in the font generation task based on diffusion models.

- Different from the image-to-image translation framework, we introduce a style encoder and a content encoder to decouple the two attributes of characters. We use the encoded condition variables as the input of the diffusion model, and guide the diffusion models to generate corresponding characters. The introduction of encoders enables our model to generalize to unseen content and unseen styles.

- We make the style encoding more stable by increasing the input of the style encoder. In order to control the quality of generated characters, we use a classifier-free diffusion model to generate. We jointly train the conditional and unconditional diffusion models and use a conditional scale to control the quality of generation in the sampling stage. Experiments have shown that our method can effectively generate Chinese character images of corresponding styles with rich details.

## 2 RELATED WORKS

### 2.1 IMAGE-TO-IMAGE TRANSLATION

The task of image-to-image translation aims to learn a mapping function that can transform a source domain image into a corresponding image that preserves the content of the original image but exhibits the desired stylistic features of the target domain. Pix2pix(Isola et al., 2017) is the first image-to-image translation model based on conditional GAN(Mirza & Osindero, 2014), the method is in supervised learning and requires paired training data. However, paired training data is not available in many cases. To achieve unsupervised image-to-image translation, many works have been proposed, in which CycleGAN(Zhu et al., 2017) introduces cycle consistency between source and target domains to discover sample relations between the two domains. However, the above methods can only be translated from one domain to another specific domain. To address this issue, many works(Chen et al., 2018; Liu et al., 2019; Bhattacharjee et al., 2020; Baek et al., 2021) have been proposed to achieve multi-class unsupervised image-to-image translation, capable of translating images between multiple visible classes. It is worth recalling that FUNIT(Liu et al., 2019) further extends its generalization ability to unseen classes by learning to encode content images and class images separately and combine them with AdaIN(Huang & Belongie, 2017).

### 2.2 UNSUPERVISED FEW-SHOT FONT GENERATION

Few-shot font generation aims to generate a new font library in the required style using only a few reference images as input. Early methods(Zi2zi; Rewrite; Lyu et al., 2017; Chang et al., 2018) regard the FFG task as an image-to-image translation problem since both tasks learn a mapping from a source domain to a target domain. These structures limit the ability of the model to generalize to unseen fonts."Zi2zi" (Zi2zi) and "Rewrite" (Rewrite) implement font generation of thousands of character pairs based on GAN(Goodfellow et al., 2020) for strong supervision. After that, based on zi2zi (Zi2zi), a series of models to improve the generation quality are proposed. PEGAN (Sun et al., 2018) builds a multi-scale image pyramid that conveys information through refined connections. DC-Font(Jiang et al., 2017) introduces a style classifier to obtain a better style representation. However, all the above methods are performed in supervised learning and require a large amount of paired data.

Unsupervised font generation does not require paired data and learns how to decouple style and content. SA-VAE(Sun et al., 2017) and EMD (Zhang et al., 2018b) disentangle the representations of style and content and can generate images for all style-content combinations. Subsequent methods(Huang et al., 2020; Wu et al., 2020; Tang et al., 2022; Park et al., 2021b; Jiang et al., 2019; Kong et al., 2022; Park et al., 2021a) follow this approach and use the introduction of additional information to enhance the style representation, such as strokes and components. RD-GAN(Huang et al., 2020) proposes a radical extraction module for roughly extracting radicals, which can improve the performance of the discriminator for few-shot Chinese font generation. CalliGAN(Wu et al., 2020) decomposes characters into components and introduces low-level structural information such as stroke order to guide the generation process. FS-Font(Tang et al., 2022) proposes a Style Aggregation Module to learn fine-grained local styles and spatial correspondences between content and reference images. SCFont(Jiang et al., 2019) reduces missing strokes in generated images and improves generation quality by using stroke information. MX-Font(Park et al., 2021b) has a multi-head encoder that specializes in different local features in a given image in a weakly supervised manner. LF-Font(Park et al., 2021a) introduces component encoders and factorization modules to capture local details. CG-GAN (Kong et al., 2022)supervises the font generator to decouple content and style at the component level through a component-aware module. DG-Font(Xie et al., 2021) introduces a functional deformation skip connection module to replace the traditional convolution block, which achieves excellent performance without any extra labels. CF-Font(Wang et al., 2023) proposes a content fusion module that projects content features into a linear space defined by the content features of the basic font in order to improve the quality of generated characters. However, most of the font generation methods so far are based on generative adversarial networks(Goodfellow et al., 2020). The two most fatal shortcomings of GAN: difficult to train and mode collapse, limit the performance improvement of this task. Diff-font(He et al., 2022) applies the diffusion model to the font generation task for the first time. It encodes the three attributes of the content, strokes and style

as conditions to generate characters. However, because this method only uses one style reference picture as input, the style of the generated characters is not stable.

## 2.3 DIFFUSION MODELS

The diffusion model is a new type of generative model consisting of a forward process (signal to noise) and a reverse process (noise to signal). Denoising Diffusion Probability Model (DDPM) (Sohl-Dickstein et al., 2015; Ho et al., 2020) is a latent variable model in which a denoising autoencoder gradually converts Gaussian noise into a signal. Score-based generative model (Song & Ermon, 2019; 2020; Song et al., 2020b)trains a neural network to predict a score function, which is used to draw samples via Langevin Dynamics. DDIM(Song et al., 2020a) extends the original DDPM to non-Markovian cases in order to speed up the sampling process and enables accurate predictions with large step sizes. Diffusion model has demonstrated comparable and superior image quality compared to GAN while showing better pattern coverage and training stability. It has also been explored for conditional generation, such as class-conditional generation, image-guided synthesis, and super-resolution(Choi et al., 2021; Dhariwal & Nichol, 2021; Song et al., 2020b; Meng et al., 2021). A classifier-guidance mechanism(Dhariwal & Nichol, 2021) is proposed, which uses pre-trained classifiers to provide gradient guidance for generating images of target classes. This method requires an additional classifier to be trained and has high-performance requirements for the classifier. Then a classifier-free guidance diffusion model(Ho & Salimans, 2022) is proposed, which achieves the effect of generating target categories by jointly training the conditional and unconditional diffusion models. ILVR (Choi et al., 2021) proposes a way to iteratively inject image guidance to a diffusion model, but it exhibits limited structural diversity of the generated images. DALL-E2(Ramesh et al., 2022) and Imagen(Saharia et al., 2022) introduce a pre-trained text encoder to generate semantic latent spaces and achieve remarkable results in text-to-image tasks. Diffusion model is thriving as a powerful generative model with diversity, training stability, and easy scalability which GAN usually lacks. In this work, we apply classifier-free diffusion guidance to our diffusion model-based font generation framework to enable the supervision of images in terms of the realism of synthetic images and consistency with character structures.

## 3 PROPOSED METHOD

The general architecture of our proposed method is shown in Fig2. It consists of a character attribute encoder and a generation framework DDPM(Ho et al., 2020). Given a content image $I_c$ and a style image $I_s$, the character attribute encoder will encode the content and style respectively, and concatenate them together as latent variables. DDPM receives latent variables as input and learns how to generate character images corresponding to content and style from Gaussian noise.

### 3.1 CHARACTER ATTRIBUTE ENCODER

The character attribute encoder $f$ consists of a content encoder $E_c$ and a style encoder $E_s$, both of which are pre-trained from DG-Font(Xie et al., 2021), and their parameters are frozen during training. The character attribute encoder has two inputs, one is the content image $I_c$ and the other is the style image $I_s$. The process can be written as:

$$z_c = E_c(I_c), \tag{1}$$

$$z_s = \frac{1}{n} \sum_{i=1}^{n} E_s(I_{s_i}), \tag{2}$$

where $n$ is the number of reference images input to the style encoder. The output of the character attribute encoder can be denoted as $z = f(z_c, z_s)$.

The input of the content image is obtained from the character image in a manually selected font (referred to as the source font). We choose the Chinese $Song$ font as the source font, which has a square structure so that the content encoder can extract it correctly to content features. Using a content encoder also allows our model to generate unseen content. Different from other methods that only use one style image as the reference style input, we use four random style images in the target style to input the style encoder. Then we average the four style encodings and concatenate it

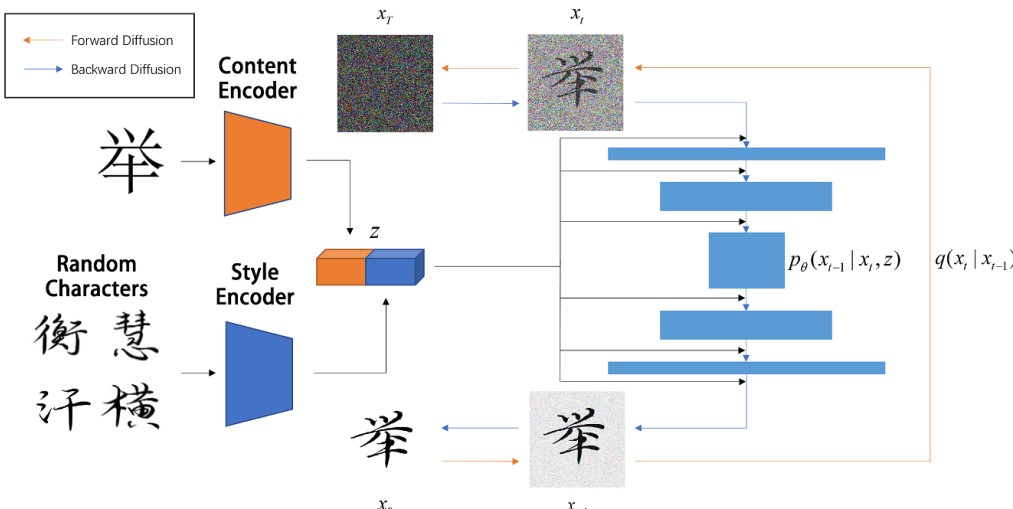

Figure 2: **The framework of our model.** (a) We first train DG-Font to obtain a style encoder and a content encoder, whose parameters are frozen during DDPM training. (b) In the forward diffusion process, we slowly add Gaussian noise to the original image, making it completely a Gaussian noise image after $T$ time steps. (c) In the reverse diffusion process, we randomly select four Chinese character images with the same style as the input of the style encoder, and use the corresponding source font image of the content as the input of the content encoder to obtain a latent variable $z$, which is used as a conditional input to DDPM to predict the noise added at each step in the forward diffusion process.

with content encoding before inputting them to DDPM. This is because we believe that it is difficult to accurately extract the font style only by relying on a style image, especially if the one has fewer strokes. If the number of input style images is increased, the style features encoded by the style encoder will be closer to the real style feature.

## 3.2 DIFFUSION PROCESS

Our model uses DDPM(Ho et al., 2020) as the generator. The diffusion process is divided into two stages, called forward diffusion and backward diffusion. We take the original ground-truth images determined by the two attributes of content and style as samples in the training data distribution, denoted as $x_0 \sim q(x_0)$.

In the process of forward diffusion, we slowly add random Gaussian noise $\mathcal{N}$ to the real image $x_0$ in a total of $T$ time steps, just like the thermal motion of molecules, gradually transforming it from a stable state to a chaotic state. Then we obtain the real image $x_0$ from Long Markov chains to isotropic Gaussian noise images $x_T$, which can be written as:

$$q(x_{1:T}|x_0) := \prod_{t=1}^{T} q(x_t|x_{t-1}), \tag{3}$$

where

$$q(x_t|x_{t-1}) = \mathcal{N}(x_t; \sqrt{1-\beta_t}x_{t-1}, \beta_t \mathrm{I}), \tag{4}$$

$t \in \{1, ..., T\}$ and $\beta_t$ is a predefined parameter used to control the mean and variance of the noise following Ho et al. (2020). According to Eq.3 and Eq.4, $x_t$ can be denoted as:

$$x_t = \sqrt{\bar{\alpha}_t}x_0 + \sqrt{1-\bar{\alpha}}\epsilon_t \sim \mathcal{N}(x_t; \sqrt{\bar{\alpha}_t}x_0, \sqrt{1-\bar{\alpha}_t}\mathrm{I}), \tag{5}$$

where $\alpha_t = 1 - \beta_t$ and $\bar{\alpha}_t = \prod_{s=1}^{t} \alpha_s$. We can clearly infer that when $T \to \infty$, $\alpha_t$ is close to 0 , $x_T$ almost obeys $\mathcal{N}(0, \mathrm{I})$, and the posterior $q(x_{t-1}|x_t)$ is also a Gaussian from Eq.5.

In the process of backward diffusion, we need to denoise the isotropic Gaussian noise $x_T$ from the latent variable $z$ obtained by the character attribute encoder to generate new data. It requires calculating the posterior $q(x_{t-1}|x_t)$ on the entire data set, which is computationally infeasible, so we approximate the posterior distribution $q(x_{t-1}|x_t)$ by a distribution $p_\theta(x_{t-1}|x_t)$ modeled by a trainable neural network, which can be written as:

$$p_\theta(x_{t-1}|x_t, z) = \mathcal{N}(\mu_\theta(x_t, t, z), \sum\nolimits_\theta (x_t, t, z)), \tag{6}$$

where $\mu_\theta$ and $\sum_\theta$ predict the mean and variance of the noise respectively. To simplify the model, we usually fix the variance as a constant and train the model to learn the mean. To recover the denoised data, we subtract the estimated noise $\epsilon_\theta(x_t, t)$ at time $t$ from the noisy data, which can be denoted as:

$$p_\theta(x_{0:T}, z) := p(x_T) \prod_{t=1}^{T} p_\theta(x_{t-1}|x_t, z) \tag{7}$$

This diffusion process is learned by a modified U-Net model introduced in Dhariwal & Nichol (2021), which augments the original architecture with self-attention between intermediate layers.

We optimized the diffusion model with a standard MSE loss between the estimated noise and the noise added in the forward diffusion process, which can be written as:

$$L_{simple}(\theta) = \mathbb{E}_{x_0, \epsilon, z} \left[ \|\epsilon - \epsilon_\theta (x_t, t, z)\|^2 \right], \tag{8}$$

where $\epsilon_t$ is the noise added at the timestep $t$ in the forward diffusion process, while $\epsilon_\theta(x_t, t)$ is the noise estimated by the U-Net model at time $t$. In this way, we can use the latent variable $z$ from the character attribute encoder to generate character images from Gaussian noise.

### 3.3 CLASSIFIER-FREE GUIDANCE DIFFUSION

To achieve classifier-free guided generation, we employ a special training strategy. We randomly discard content attribute vectors or style attribute vectors with a probability of 30%. Such a strategy can jointly train conditional and unconditional diffusion models to achieve the role of classifier-free guided generation. When sampling, we convert the prediction noise $\epsilon_\theta$ to:

$$\begin{aligned}
\hat{\epsilon}_\theta (x_t, t, f(z_c, z_s)) &= \epsilon_\theta(x_t, t, 0) \\
&+ s * (\epsilon_\theta (x_t, t, f(z_c, z_s)) - \epsilon_\theta(x_t, t, 0)),
\end{aligned} \tag{9}$$

where 0 is the discarded attribute vectors and s is the guidance scales of content and styles.

In this way, we can obtain $x_{t-1}$ from $x_t$ as follows:

$$x_{t-1} = \frac{1}{\sqrt{\alpha_t}} \left( x_t - \frac{\beta_t}{\sqrt{1 - \bar{\alpha}_t}} \hat{\epsilon}_\theta \right) + \sqrt{1 - \alpha_t} z, \tag{10}$$

where $z \sim \mathcal{N}(0, \mathrm{I})$. By iterating Eqn. 10, the final character image $x_0$ with the given style and content can be obtained.

## 4 EXPERIMENTAL RESULTS

In this section, we evaluate our proposed method with the Chinese font generation task. We first introduce our dataset. After that, we show our experimental results comparing our proposed method with other previous advanced methods quantitatively and qualitatively.

### 4.1 EXPERIMENTAL SETTINGS

**Datasets.** To evaluate our model for Chinese font generation, we collect a dataset that contains 400 fonts including handwritten fonts and printed fonts, each of which has 1000 commonly used Chinese characters. The training set contains 360 fonts, and each font contains 800 characters randomly chosen from 1000 characters. The testing set consists of two parts. One part is the remaining 200 characters of the 360 fonts. Another part is the remaining 40 fonts and each of them contains the

Figure 3: **Generated results of each methods on testing dataset.** We mark erroneous skeletons and other mismatched styles with boxes.

| Methods | RMSE↓ | SSIM↑ | LPIPS↓ | FID↓ |
|---|---|---|---|---|
| | Seen fonts | | | |
| FUNIT | 0.3179 | 0.5735 | 0.1544 | **22.1452** |
| MX-Font | 0.3093 | 0.5749 | 0.1345 | 34.4167 |
| DG-Font | 0.2969 | 0.5878 | 0.1266 | 42.3316 |
| CG-GAN | 0.2889 | 0.5959 | 0.1243 | 35.9291 |
| CF-Font | **0.2821** | **0.6045** | **0.1166** | 38.2691 |
| Ours | 0.2883 | 0.5962 | 0.1252 | 28.5714 |
| | Unseen fonts | | | |
| FUNIT | 0.3446 | 0.5435 | 0.1795 | 22.3647 |
| MX-Font | 0.3247 | 0.5597 | 0.1574 | 26.6849 |
| DG-Font | 0.3137 | 0.5636 | 0.1434 | 28.4021 |
| CG-GAN | 0.3004 | 0.5712 | 0.1465 | 23.4247 |
| CF-Font | **0.2984** | **0.5759** | **0.1401** | 34.9543 |
| Ours | 0.3101 | 0.5650 | 0.1494 | **19.5461** |

Table 1: **Quantitative evaluation on the whole dataset.** We evaluate the methods with state-of-the-art methods on seen/unseen fonts. Bold and underlined numbers denote the best and the second best respectively.

whole 1000 characters, which are used to test the generation performance of unseen fonts. In order to make a fair comparison with the prior methods, the image size is uniformly set as $80 \times 80$.

**Evaluation Metrics.** We use several metrics in image generation methods for quantitatively comparing our method with other advanced methods, *e.g.*, RMSE, SSIM (Wang et al., 2004), LPIPS (Zhang et al., 2018a), FID (Heusel et al., 2017). RMSE and SSIM are pixel-level metrics, while LPIPS and FID are perceptual-level metrics. RMSE (Root Mean Square) is used to measure the

similarity between the generated image and the real image at the pixel level by calculating the mean square error. SSIM(Structural Similarity) measures the structural similarity of two pictures through three features of the picture: luminance, contrast, and structure, which is more in line with the intuitive feeling of human eyes. LPIPS (Learned Perceptual Image Patch Similarity) measures the distance between two images in deep feature space. FID (Fréchet Inception Distance) is a measure used to calculate the distance of the feature vector between the real image and the generated image.

## 4.2 IMPLEMENTATION DETAILS

We train our model using Adam(Kingma & Ba, 2014) with $\beta_1 = 0.9$ and $\beta_2 = 0.99$ for the style encoder, and RMSprop with $\alpha = 0.99$ for the content encoder. The learning rate and weight decay are both set as $10^{-4}$. We trained our model for 800,000 iterations with a batch size of 32 and got a model with stable final results.

In the inference stage, we use Song, a font commonly used in font generation tasks, as the content font for fair comparison. In order to unify the image size to $80 \times 80$, we have slightly modified the network structure of several methods to adapt to the size of the input image and the few-shot setting. In our method, we set the guidance scale to 2.0, which enables the model to generate stable content and style. At the same time, we adopt DDIM(Song et al., 2020a) and compress the time step to 25 to speed up the generation process.

## 4.3 COMPARISON WITH STATE-OF-THE-ART METHODS

In this section, we compare our model with five state-of-the-art methods, including an image-to-image translation method (FUNIT(Liu et al., 2019)), two component-related methods (MXFont(Park et al., 2021b), CG-GAN(Kong et al., 2022)), DG-Font(Xie et al., 2021) and CF-Font(Wang et al., 2023).

**Quantitative comparison.** Table. 1 shows the results of a quantitative comparison between our method and other previous state-of-the-art methods. On the seen fonts, our method is suboptimal in RMSE and SSIM metrics and is basically on par with DG-Font and CG-GAN in LPIPS metrics. This is because the results of the diffusion model in generating tasks are not stable enough. Once the model is determined, the generation results of GANs are determined, while diffusion requires multiple generations to select better results. For fonts that have not been seen before, our method has decreased in metrics and is generally in the third position in various metrics. It is speculated that the reason may be the insufficient generalization ability of the pre-trained style encoder. For FID metric, although FUNIT performs well, it performs too poorly on other metrics.

**Qualitative comparison.** The qualitative results are shown in Fig. 3. Our reference characters are randomly selected, including both seen and unseen content. The characters we generate are of high quality in terms of style consistency and structural correctness. The results of FUNIT often have structural errors and incomplete strokes, especially on fonts that have never been seen before. MX-Font can maintain the shape of characters to a certain extent, but it often produces blurry characters and stroke adhesion. The generation of DG-Font and CG-GAN is generally good, but there are noise and certain style shifts, and there are also missing details. CF-Font generates well on most fonts, but has limitations for complex content and difficult style generation. Our proposed method can generate high-quality character images in both test sets, which retain accurate information about the target style. Even for difficult styles, the corresponding style can be accurately generated. For complex content, it is also possible to retain complete stroke information.

Due to the particularity of font generation tasks, which involve issues such as consistent style and complete strokes, some quantitative metrics cannot fully reflect the generation ability of the model. In qualitative evaluation, our method produces images with significantly better visual quality than other methods.

## 4.4 ABLATION STUDIES

In this section, we further discuss the impact of the number $n$ of input images to the style encoder with some ablation studies. Except for the number of images for the input style parameter, other experimental settings are consistent. All results are tested on both parts of the test set.

| $n$ | RMSE↓ | SSIM↑ | LPIPS↓ | FID↓ |
|---|---|---|---|---|
| Seen fonts | | | | |
| $n = 1$ | 0.3032 | 0.5821 | 0.1442 | 32.2457 |
| $n = 4$ | 0.2883 | 0.5962 | 0.1252 | 28.5714 |
| Unseen fonts | | | | |
| $n = 1$ | 0.3142 | 0.5596 | 0.1542 | 34.3168 |
| $n = 4$ | 0.3101 | 0.5650 | 0.1494 | 19.5461 |

Table 2: Effectiveness of increasing the number of style reference images.

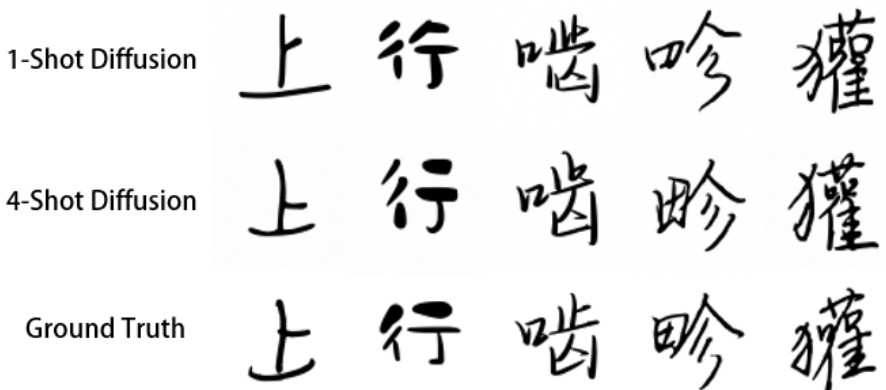

Figure 4: **Qualitative results on ablation study.** Some error generation results using one style reference character are presented, including inconsistent styles, excessive strokes and missing structures.

We train two models separately, one using only one style reference image and the other using 4 style reference images. The reason for choosing 4 instead of other numbers is a choice after weighing the quality of generation and training overhead. As shown in Table. 2, when adding the number of style reference pictures, the quantitative results improve in all evaluation metrics, which shows the effectiveness of our improvement. According to Fig.4, in the case of generating characters with difficult structures, characters generated from multiple style reference pictures can ensure the correctness and uniformity of style. The font generated by the model that only uses one style reference image has shortcomings such as style deviation, redundant strokes and incomplete structure.

## 5 CONCLUSION

In this paper, we propose an efficient model for unsupervised font generation based on the diffusion model, which is able to generate realistic characters without paired images and generalizes well to unseen fonts. We use a character attribute encoder to encode two attributes of Chinese characters: content and style. To ensure the accuracy of the generated character styles, we increase the number of input style reference images. This is one of the few applications of diffusion models in font generation tasks. A large number of Chinese font generation experiments verify the effectiveness of the diffusion models.

**Limitations.** Due to our method being based on the diffusion models, it has the problem of slow inference speed. Although we have adopted methods such as DDIM to accelerate the generation process, it is still slower compared to GAN-based methods. In addition, due to the random Gaussian noise in the sampling process of the diffusion model, each generation result will be different, which requires manual judgment and regeneration of the generated incorrect images. We will also explore better methods to control the stable generation of diffusion models in the future.

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
