# OpenReview forum: "Attribute-Guided Diffusion for Unsupervised Few-Shot Font Generation"
_ICLR.cc/2024/Conference — ICLR 2024 Conference Withdrawn Submission_

### Official Review · Reviewer_N8n1 · 2023-10-30

**Soundness:** 2 fair
**Presentation:** 3 good
**Contribution:** 2 fair
**Rating:** 5
**Confidence:** 4

**Summary:**

This paper proposes an unsupervised few-shot font generation method using diffusion models. Specifically, the authors use a character attribute encoder to extract content and style vectors of glyphs. Then, the encoded vectors are used to condition the diffusion model to generate fonts with desired content and style.

**Strengths:**

1. The manuscript is well-organized and easy to follow.
2. From the qualitative results, the proposed method is superior to other few-shot font generation methods.

**Weaknesses:**

1. The novelty of this paper is limited. Currently, Diff-Font [1] has utilized the diffusion model to achieve few-shot font generation task. In addition, some techniques (such as classifier free guidance) and networks have been widely applied in the field of image generation.

2. From Table 1, it can be seen that the proposed method is not superior to the existing few-shot font generation methods in quantitative evaluation. In addition, I suggest that the author add some comparative methods in the experiment, such as Diff-Font [1], MF-Net [2] and VQ-font [3].


[1] He, H., Chen, X., Wang, C., Liu, J., Du, B., Tao, D., & Qiao, Y. (2022). Diff-Font: Diffusion Model for Robust One-Shot Font Generation. arXiv preprint arXiv:2212.05895.

[2] Zhang, Y., Man, J., & Sun, P. (2022, October). MF-Net: a novel few-shot stylized multilingual font generation method. In Proceedings of the 30th ACM International Conference on Multimedia (pp. 2088-2096).

[3] Pan, W., Zhu, A., Zhou, X., Iwana, B. K., & Li, S. (2023). Few shot font generation via transferring similarity guided global style and quantization local style. In Proceedings of the IEEE/CVF International Conference on Computer Vision (pp. 19506-19516).

**Questions:**

This work seems to be a combination of some existing work. Are there any special designs made for scenarios with few samples?

---

> ### Author Response · Authors · 2023-11-14
>
> Thank you for your review! Diff-Font only uses one style reference image as style input, and we think this cannot fully extract style features. We make full use of few samples as style input. As for Table 1, since the results generated by diffusion models are different each time, the quantitative comparison has some fluctuations. We will also continue to explore ways to improve the stability and quality of model generation. Notably, Diff-Font has not yet been published, which means its effectiveness is still unknown.

---

### Official Review · Reviewer_28JC · 2023-10-30

**Soundness:** 2 fair
**Presentation:** 2 fair
**Contribution:** 3 good
**Rating:** 5
**Confidence:** 4

**Summary:**

The paper offers an approach to font generation, using the diffusion model to address the issues of training instability and model collapse commonly associated with traditional GAN methods. The method demonstrates stable training on large datasets and impressive qualitative and quantitative results. It is considered a significant contribution to the field, given the importance of the font generation problem. However, the paper needs to provide more detailed explanations for the addressed problems, better describe where innovation lies within its components, and clarify the implications of comparative results, particularly the substantial decrease in FID.

**Strengths:**

1. The paper introduces a novel generative model to font generation field, the diffusion model, which effectively addresses the issues of training instability and model collapse that traditional GAN methods often face.

2. The proposed method demonstrates the ability to achieve stable training on large-scale datasets, which is a significant achievement in the field of font generation.

3. The paper showcases impressive results in both qualitative and quantitative analyses, highlighting the potential effectiveness of the method.

4. The paper tackles an important problem in font generation, and its "Attribute-Guided Diffusion" approach is considered a significant contribution to the field.

**Weaknesses:**

1. Lack of Detailed Explanation: The paper lacks a comprehensive explanation of the root causes of training instability and model collapse in traditional GANs and how the proposed diffusion model-based framework effectively addresses these issues. More detailed explanations and experiments are needed.

2. Insufficient Innovation Description: The three components of the method appear to build on existing models, but the paper doesn't sufficiently clarify where innovation lies within these components or how they improve upon existing approaches.

3. Inconclusive Comparisons: Comparative analysis with other methods, especially FUNIT, does not clearly demonstrate a substantial advantage in terms of visual perception. The paper compares its method to the FUNIT method in 2019, but the advantages of the proposed approach over FUNIT do not seem very pronounced in terms of visual perception. Similar issues exist when comparing the method to other approaches. While improvements are noted in terms of RMSE, SSIM, and LPIPS, there is a significant decrease in FID. It would be helpful to explain the reasons behind this change and whether it signifies the effectiveness of the proposed method.

**Questions:**

1. What are the root causes of training instability and model collapse in traditional GANs, and how does the proposed diffusion model-based framework mitigate these issues? Can this be supported with additional experimental evidence?

2. Where specifically does the paper introduce innovation within the three method components (Character Attribute Encoder, Diffusion Process, and Classifier-Free Guidance Diffusion)?

3. Could the paper provide more context and explanation regarding the noticeable decrease in FID when compared to other methods? What does this imply for the overall effectiveness of the proposed approach?

---

> ### Author Response · Authors · 2023-11-14
>
> Thank you for your review! My reply is below:
> 1. The training instability of GANs comes from its use of adversarial loss, which requires training two networks at the same time. In order to fool the discriminator, the generated results after the generator training may be lack of diversity and cannot cover all sample distributions. This leads to model collapse. The diffusion model has a simple loss and only needs to train a network, which can effectively solve the problems existing in GANs. Relevant experimental results have been described in some previous works, such as the paper named Diffusion Models Beat GANs on Image Synthesis.
> 2. In the character attribute encoder, we improve the accuracy of target style extraction by increasing the input reference image of the style encoder. In order to improve the guidance of condition variables, we introduce Classifier-Free Guidance Diffusion to obtain better generation quality.
> 3. The performance of the FID indicator is indeed a bit strange. We are not sure whether FID is suitable for evaluating font generation tasks (grayscale, pictures with obvious boundaries). We and previous papers have more or less mentioned this. FID is sometimes ambiguous with other indicators. There will be situations where some methods have poor indicators but high FID, such as DG-Font and CF-Font. Since diffusion models generate different results every time, the quantitative results have many instabilities. We prefer to evaluating our methods through qualitative results.

---

### Official Review · Reviewer_BnxP · 2023-11-01

**Soundness:** 2 fair
**Presentation:** 2 fair
**Contribution:** 1 poor
**Rating:** 3
**Confidence:** 4

**Summary:**

This paper proposes an attribute-guided diffusion model for font generation.
The network takes both content font and style font as input and encodes them into two latent codes, which are taken as the diffusion model to generate font with input content in a new style.

**Strengths:**

The paper is clear and easy to follow.
Both qualitative and quantitative results are shown to prove the effectiveness of the proposed method.
It is a good attempt to apply the diffusion model to the font generation task.
Using few shot might increase the stability of the font style transfer results.

**Weaknesses:**

Miss comparison with recent `Diff-Font: Diffusion Model for Robust One-Shot Font Generation`.
The idea of Diff-Font is very close to the proposed method though it only uses one style image as a reference.

The performance does not overpass the SOTA.
I do not see any advantage of the proposed method in quantitative results in Table 1 (compared to CF-Font).
Is there any reason for that?

One important thing about the font is usually vector graphics.
Is there any possibility to convert the image into a vector representation?

**Questions:**

As mentioned in the weakness part, my first question is on the performance level.
Compared to CF-Font, it does not provide any benefit to the font generation task.
Is there any reason for that?

To my understanding, the difference between Diff-Font and this paper is that this paper uses multiple style font references.
I am wondering if this paper can show some single-style reference test results.

The quantitative results are worse than the current SOTA, is there any reason for this?
Is it possible to show more qualitative results to better evaluate this pipeline?

---

> ### Author Response · Authors · 2023-11-14
>
> Thank you for your review!
>
> As for the first question, since diffusion models generate different results every time, the quantitative results have many instabilities. But in terms of visual quality, we are able to match the best methods, and designers can choose the one with better quality through multiple generations. At the same time, this is also one of the few explorations of diffusion models in the field of font generation.
>
> To answer the second question, we also tried comparing ours with Diff-Font. Since it only uses one style reference image, the generated style is extremely unstable. It takes a lot of time to generate multiple selections in qualitative comparisons.
>
> For the last quesion, as mentioned above, due to the instability of diffusion models, there may be some disadvantages in quantitative comparison, but this also provides designers with the opportunity to generate multiple times and then choose the better one.
>
> Notably, Diff-Font has not yet been published, which means its effectiveness is still unknown.

---

### Official Review · Reviewer_xmu4 · 2023-11-02

**Soundness:** 1 poor
**Presentation:** 3 good
**Contribution:** 2 fair
**Rating:** 3
**Confidence:** 5

**Summary:**

This paper describes an unsupervised few-shot font generation method based on diffusion model. Qualitative and quantitative results show the usefulness of the approach.

**Strengths:**

The paper is overall well-written and easy to follow.

**Weaknesses:**

1. The paper, while offering insights, seems to lack substantial novelty. Its resemblance to the Diff-Font framework is notable, yet the paper does not adequately delineate the differences between its approach and that of Diff-Font.
2. The experimental evaluation appears to be somewhat limited. The paper mentions over 60,000 Chinese characters, yet the evaluation only encompasses 1,000 characters. This selection might not fully represent the entire scope, potentially affecting the convincingness of the results. Moreover, there seems to be an absence of a significant evaluation metric: user studying. This metric is notably present in prior research, for instance in works like MX-Font, CG-GAN, CF-font, etc. Additionally, the evaluation focuses solely on Chinese characters. Although the paper acknowledges the intricacy of other languages such as Korean and Japanese, which also possess rich characters and complex structures, these languages have not been included in the evaluation. This exclusion could limit the comprehensiveness of the study's findings.
3. The method's performance in quantitative comparison appears to be less effective than that of CF-Font.

**Questions:**

This paper appears to have limited novelty due to a lack of substantial original design elements. To enhance its innovative aspects, the authors might consider improvements in model design, particularly focusing on the interaction between the attribute encoder and the diffusion model. As it stands, the current version falls short of meeting the standards set by ICLR.

Furthermore, while the authors effectively highlight the limitations of other methods, they do not provide examples of failure cases for their own approach. This omission is particularly noticeable given that the qualitative results presented seem superior. It is surprising that these promising qualitative outcomes are not reflected in superior quantitative results when compared to other methods. The authors are encouraged to identify and discuss the weaknesses of their proposed method. Doing so could offer valuable insights and direct attention towards potential areas for future improvement.

---

> ### Author Response · Authors · 2023-11-14
>
> Thank you for your review! Since diffusion models generate different results every time, the quantitative results have many instabilities. But this also provides designers with the opportunity to generate multiple times and then choose the better one. This is also one of the few explorations of diffusion models in the field of font generation. We prefer to evaluating our methods through qualitative results. Due to the slow sampling speed of diffusion models, we did not evaluate all Chinese character datasets, but we evaluated a large number of Chinese characters commonly used. We will also continue to innovate our models to improve the stability and quality of generation. Notably, Diff-Font has not yet been published, which means its effectiveness is still unknown.